# A Path Planning Method for Autonomous Vehicles Based on Risk Assessment

**Wei Yang, Cong Li \* and Yipeng Zhou**

School of Mechanical and Automotive Engineering, Shanghai University of Engineering Science,
Shanghai 201620, China

\* Correspondence: licong@sues.edu.cn

**Abstract:** In order to meet the requirements of vehicle automatic obstacle avoidance, a lane change trajectory planning method is proposed to meet the requirements of safety, comfort, and lane change efficiency. Firstly, the potential collision points that may exist are analyzed using information about surrounding vehicle movement and the road. Then, the safe lane change range for vehicles is obtained. Secondly, the control points of the fifth order Bézier curve are constrained to generate a series of path clusters in the optimal range. At the same time, the driver's style and reaction time are taken into account in the risk assessment stage of the route using the improved artificial potential field method. Finally, the optimal path is selected by comprehensively considering lane-changing efficiency and comfort. In order to further verify the accuracy of the algorithm, real-vehicle experiments have been carried out on the autonomous vehicle platform. Under different driving styles, the vehicle can avoid obstacles perfectly while ensuring the smoothness of the path. Simulation and real-vehicle experiment results show that the proposed algorithm can provide an excellent solution for autonomous vehicles for lane changing and obstacle avoidance.

**Keywords:** autonomous vehicles; obstacle avoidance; path planning; risk assessment; potential collision points

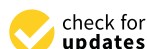



## 1. Introduction

Nowadays, it is estimated that most traffic accidents are due to human errors. In particular, accidents are more likely to occur when vehicles are changing lanes to avoid obstacles [1]. With more and more scientists and R&D teams focusing on the driverless industry, the entire perception system and control technology have developed rapidly, and autonomous vehicles have attracted extensive attention from all walks of life for reducing driver fatigue and improving driving safety [2]. The driverless vehicle was first proposed by Norman Bel Geddes, an American designer. At present, many companies are actively promoting their own driverless solutions, such as Google and Tesla. The entire architecture of autonomous vehicles includes environmental perception, decision planning [3–5], and executive control [6,7]. The vehicle obtains surrounding environmental information through sensors and high-precision maps. The decision-making module is in charge of generating high-level orders, such as slow down and speed up, and the planning module is to compute the detailed trajectory profile [8]. Finally, the control module is executed to control the next action of the vehicle.

This paper focuses on local path-planning methods that are led by global path information, such as the potential field method, search sampling algorithm, and discrete optimization algorithm, which are all widely used approaches for local path planning. The artificial potential field method establishes a repulsive potential field around the obstacle and a gravitational potential field around the target point. The controlled object is subjected to repulsion and gravitational action in the compound field composed of these two potential fields, and the combined force guides the movement of the controlled object and searches

for a collision-free obstacle-avoidance path. The artificial potential field is a relatively mature algorithm that is widely used for its math calculations [9]. However, there is the problem of local minimum and unreachable target in this algorithm. Ma et al. proposed improving the gravitational potential field model [10], using logarithmic function instead of quadratic function. The improved model reduces the variation range of the potential field strength, and at the same time introduces the relative position and relative velocity to reduce the repulsive potential field strength. It solves the problem that obstacles near the target are unreachable. The regulation mechanism of random perturbation potential energy is introduced to solve the problem that it is difficult to move at local minimum points. Szczepanski et al. proposed an algorithm that uses augmented reality to bypass obstacles [11]. It uses the detection of the local minimum algorithm based on LiDAR's data as a trigger to create a wall to prevent the vehicle from falling into local minima. However, this method does not take into account whether the path is optimal or not, while only testing under stationary obstacles. The difference is that we select the optimal path satisfying the smoothness condition among multiple trajectories on the premise of ensuring safety. A better artificial potential field technique [12] is proposed by changing the obstacle repulsive field function. The artificial potential field approach is developed based on the safety model, and the potential field of obstacles is improved to an ellipse. The generated path might be able to meet road, dynamics, and kinematics requirements. This strategy, on the other hand, will lead the vehicle to turn unnecessarily. Hu et al. proposed a combined artificial potential field model considering vehicle speed [13], which includes five components, target potential, road potential, lane potential, vehicle potential, and velocity potential. The collision avoidance path for autonomous driving is calculated with gradient method from the superposition of disparate potential function. Sun et al. presented a hybrid motion planning approach based on the timed-elastic-band approach and artificial potential field [14]. It established different potential fields and the velocity of the vehicle is planned by using the conversion function of the virtual potential energy of the superimposed potential field and the virtual kinetic energy of the vehicle. The planned path has excellent obstacle avoidance in static and dynamic environments.

Search and sampling algorithms mainly include A-Star algorithm and Rapid-exploration Random Tree (RRT) algorithm. A-star is a heuristic search algorithm that finds the point with the minimum cost by traversing the surrounding nodes. The target point can be reached by searching the next node [15]. However, there are too many unnecessary turning points in the path. Zhang et al. proposed the method of A-Star combined with artificial potential field [16]. This method adds the influence of surrounding obstacles and target points on the controlled object to the estimated cost, which effectively reduces the number of turning points. The reference [17] proposed a guideline-based A-Star algorithm by setting guidelines to improve the traditional heuristic function, while using key points except obstacles to guide the planned path to avoid obstacles earlier. This type of algorithm is mainly applied in local path planning for mobile robots, which cannot satisfy the smoothness of trajectory. RRT is an efficient planning method in multidimensional space; however, it takes a long time to compute. At present, the Bidirectional Rapid-exploration Random Tree (B-RRT) algorithm is proposed [18]. Compared with the original RRT, the algorithm builds a second tree in the target area for expansion. The balance of two trees must be considered in each iteration. The two trees continue to expand towards each other alternately instead of random expansion, which makes the double tree RRT algorithm more effective than the single tree RRT algorithm. Zhang et al. proposed an improved RRT algorithm [19] which avoids over-searching by reducing the randomness of sampling points and combines the comprehensive criteria of angle and distance to improve the smoothness of the path. It also uses secure ellipse method to avoid the collision. However, the distance of the semi-major axle only considers the speed of the host vehicle. We add the speed ratio of the obstacle vehicle to the host vehicle, and the driver's style factors, as the distance of the semi-major axle. However, the path planned in this way may not be the shortest path. The search algorithm can find a feasible trajectory quickly, but the disadvantage of this method is

that the generated trajectory curvature is discontinuous. However, there are too many nodes and spikes in this path-planning process. It needs to be fitted with other methods for smoothness, such as Bézier curves [20–22], splines, and so on.

Optimization-based path-planning method uses a parametric cubic spline to define the parametric curve of the base frame that is constructed based on waypoints, and it generates a series of candidate paths in the frenet coordinate system. Finally, it introduces different evaluation indicators to filter out the best path that meets the criteria [23,24]. Zheng et al. proposed an effective trajectory planning algorithm based on the quartic Bézier curve in the frenet coordinate system [25]. This method has collision paths when generating paths. Although the risk assessed by the potential field method and the curvature of the path are comprehensively considered when screening the path, the time of the algorithm is prolonged. On the basis of this potential field risk assessment method, Zhang et al. designed a lane-changing speed curve [26]. Taking the safety impact of surrounding obstacles on the vehicle as an additional safety constraint, Chen et al. also applied the quartic Bézier curve [27], and the curvature generating problem is further reduced to finding three suitable parameters. Optimization is adopted to find the three parameters with a given objective function. In this way, the curve obtained according to the optimized parameters can meet the optimal conditions. Similarly, Tharwat et al. proposed the CPSO algorithm to optimize the control points of the Bézier curve [28]. However, the experiments were performed using only a static environment. Zhou et al. proposed an optimal lane-changing strategy to determine the final trajectory according to several look-up tables on lane-changing performance [29]. At the same time, it meets the comfort indicators, but it is only suitable for specific roads. Although the above studies can filter out safe and collision-free trajectories, there is little discussion of path selection for driving styles.

Therefore, this paper proposes a quintic Bézier curve algorithm based on potential field risk assessment. The potential field risk assessment is considered in the generation path of the quintic Bézier curve to further ensure the safety of the path. The comfort cost function and lane change efficiency function are designed to avoid obstacles accurately, safely, and efficiently. According to the driver's driving style, it chooses the path with the best comfort or lane-changing efficiency. The purpose of this paper is to find a feasible and optimal obstacle-avoidance path for the vehicle. Therefore, the core objective of this paper is to achieve a feasible collision-free trajectory for the vehicle.

The flow chart of the path-planning algorithm is shown in Figure 1, and the paper is organized as follows: Section 1 analyzes potential collision points and optimizes the control points of the fifth order Bézier curve. Section 2 designs the paths assessment based on APF. Section 3 proposes the evaluation method for lane-changing efficiency and comfort. Section 4 verifies the proposed algorithm through simulation and experiment. Section 5 puts forward the conclusions.

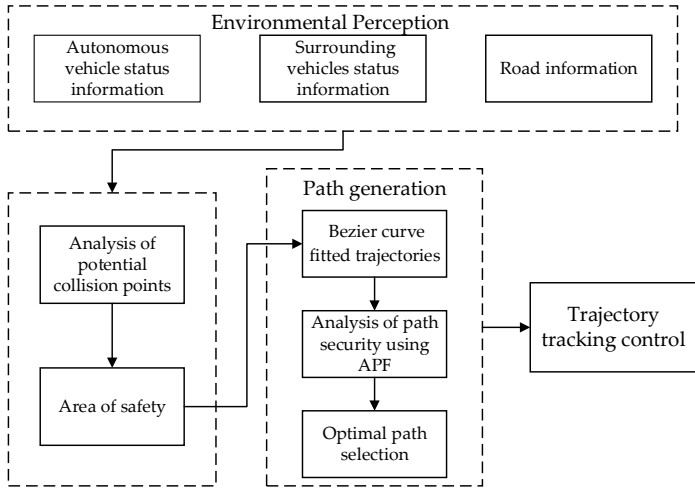

**Figure 1.** Flow chart of the path-planning algorithm.

## 2. Lane Change Curve Models

### 2.1. Analysis of Potential Collision Points of Vehicles

The premise of trajectory planning is to ensure the safety of the transition process. Figure 2 shows the potential collision scenario of vehicles during lane change. $S_1$ and $S_2$ are the critical points of collision between autonomous vehicles and vehicles in other lanes, respectively. $X_a$ and $X_f$ represent the location of autonomous vehicles when collision occurs. At the same time, the vehicle model is simplified to an elliptical model, taking the vehicle center as the origin of the ellipse, assuming that the minor axis is equal to the width of the lane, and the major axis $a$ is:

$$a = \frac{l}{2} + (1 - T_d)^{\frac{l}{w}} \frac{v_f}{v_a} \tag{1}$$

where $v_a$ is the rear vehicle speed, $v_f$ is the speed of the vehicle ahead, $T_d$ is the driver's style factor, and the aggressive style is 0.8 while the normal style is 0.2, $w$ is the vehicle width, and $l$ is the vehicle length.

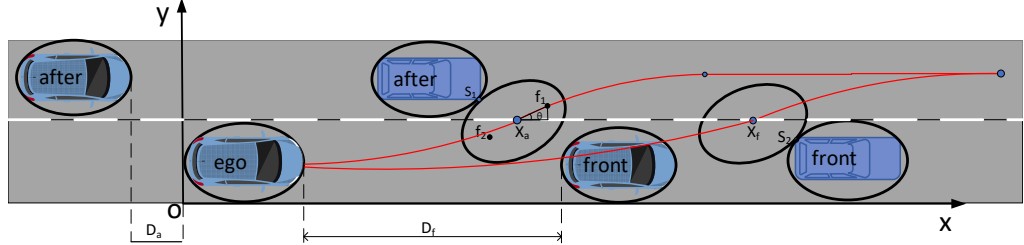

**Figure 2.** The potential collision points of vehicles.

When $v_{ego} > v_f$, the relative distance between the two vehicles will be reduced, and a collision will occur when it is less than the minimum safe distance. At this time, the collisional point is recorded as $S_2$, and the collisional time is $t_{S2}$ ($t_1 < t_{S2} < t_2$). The coordinates of $S_2$ are:

$$\begin{cases} x_{s2} = \int\limits_{t_1}^{t_{s2}} v_f dt + D_f \\ y_{s2} = \frac{L+w}{2} \end{cases} \tag{2}$$

When $v_{ego} < v_a$, the collisional point is recorded as $S_1$, and the collisional time is $t_{S1}$ ($t_1 < t_{S1} < t_2$). The coordinates of $S_1$ are:

$$\begin{cases} x_{s2} = \int\limits_{t_s}^{t_{p1}} v_1 dt - D_a \\ y_{s1} = \frac{3L-w}{2} \end{cases} \tag{3}$$

where $L$ is the lane width. According to Formulas (1)–(3), the coordinates of potential collision points can be obtained so as to determine the left and right boundaries of the lane-changing trajectory of the vehicle. As long as the vehicle position is within the boundary, and the lane change point is between $X_a$ and $X_f$, the collision can be avoided. The vehicle center coordinate $x$ is obtained according to the following formula.

$$\begin{cases} a^2 - L^2 = c^2 \\ |f_1 S_1| + |f_2 S_1| = 2a \end{cases} \tag{4}$$

$$\begin{cases} y_{f1} = c \sin \theta + L \\ x_{f1} = c \cos \theta + x \end{cases} \tag{5}$$

$$\begin{cases} y_{f2} = L - c \sin \theta \\ x_{f2} = X_a - c \cos \theta \end{cases} \tag{6}$$

where $f_1$ and $f_2$ are the focus points of ellipse, $c$ is the focal length, $\theta$ is the heading angle of the vehicle, and $|f_1 S_1|$ is the distance from $f_1$ to $S_1$.

### 2.2. Mathematical Model of Trajectory

The purpose of this section is to generate a Bézier curve cluster by constraining the control points to a feasible range. As shown in Figure 3, $P_0 = [0\ 0]'$ is the start point and $P_5$ is the end point. The first control point is the position of the vehicle itself, so it is known, and the rest of the control points are unknown during the obstacle-avoidance process.

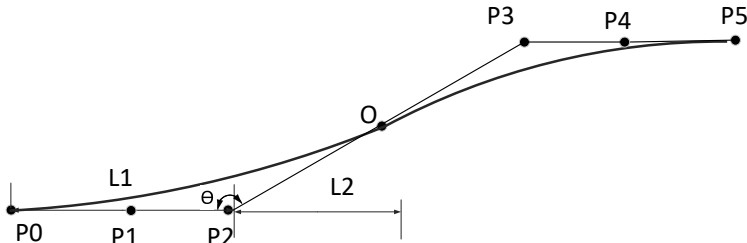

**Figure 3.** Example of quintic Bézier curve.

Therefore, in order to simplify the calculation, we make some assumptions that the lane change's trajectory should satisfy centrosymmetric. Point $O$ is the center of symmetry and also the auxiliary point, and it can control the direction during lane change. $P_1$ and $P_4$ are midpoints. The heading angle of the vehicle is always parallel to the centerline of the lane at the start and end.

Then, the connections line between $P_0$, $P_2$ and $P_3$, $P_5$ is a straight line and the value is $L_1$. Therefore, we can obtain a series of Bézier curve clusters by constraining the positions of the points $P_2$ and $P_3$. For straight line $L_1$, the vehicle is just on the center line of the lane and the heading angle $\theta$ is 0. At this time, the obstacle vehicle is directly in front of the vehicle. The constraint case is denoted as

$$d_{ac} \le d_{obs} + v_f \frac{L_1}{V_1} - (L_1 + L_b)|\cos\theta| \tag{7}$$

where $d_{obs}$ refers to the longitudinal distance between the vehicle and the front vehicle, and $L_b$ is the distance from the center of mass of the vehicle to the front of the vehicle. Therefore, under the constraint of $L_1$, it can be obtained that $P_2 = [0, L_1]'$. The coordinates of point $O$ can be expressed as $(L1 + L2, y_0)$, $X_a < L_1 + L_2 < X_f$. The curvature at point O is obtained as

$$K_\text{o} = \frac{3y_o}{4L_1{}^2} \tag{8}$$

Thus, $O = [L_1 + L_2, 4k_o L^2{}_1/3]'$. The operation between $L_1$ and $L_2$ is

$$L_2 = \frac{4k_o L_1{}^2}{3\tan\theta} \tag{9}$$

At the same time, we consider the handling stability of control points as constraint $y_o \le 0.5gt^2$. Then, we can get all the control points of the Bézier curve, such as $P_3 = [L_1 + L_2, 8k_o L^2{}_1/3]'$, $P_5 = [2L_1 + 2L_2, 8k_o L^2{}_1/3]'$, $P_1 = [0, L_1/2]$, and $P4 = [3L_1/2 + 2L_2, 8k_o L^2{}_1/3]'$.

Finally, we get all the control points of the fifth order Bézier curve. According to the Formulas (7)–(9), and the solution of the potential collision point, we can get the lane change trajectory cluster, as shown in Figure 4. In this way, the trajectories generated by the control points within the limit are feasible. Thus, the time of the algorithm is reduced, and it is not wasted on generating infeasible trajectories.

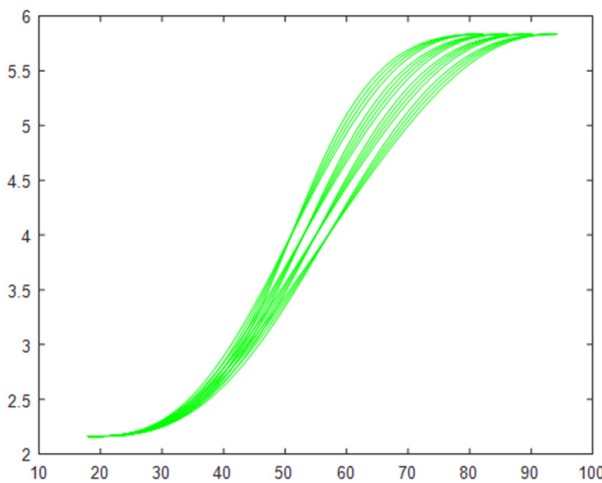

**Figure 4.** The trajectory cluster of curves.

### 3. Risk Assessment Process

Artificial potential field (APF) method has recently been widely used in the field of path planning, where it has the advantages of being easy to express in mathematical formulas, low computational effort, and high execution efficiency. However, due to the problems of local optimization and unreachable goals, the smooth path required by the vehicle cannot be planned. The traditional safety assessment uses the minimum safety distance model, which cannot assess the risk of collision with the road. At the same time, the reference [25] also uses APF as a path-security assessment. We introduce the driver's reaction time to make the model more accurate when the distance from the obstacle is less than the optimal distance. Another difference is that the attraction model is added to the road centerline, so that the vehicle can maintain a straight line. However, the attraction model does not participate in the process of risk assessment.

The sum of the road boundary potential field and the surrounding obstacle potential field are used as an evaluation method for driving safety in this paper. According to the risk assessment information, candidate paths are filtered out.

#### 3.1. Establishing Road Risk Assessment Model

Different from other similar papers, this paper adds an attractive potential field at the centerline of the lane; when there is no obstacle in front of the vehicle, the vehicle can be kept on the centerline under the action of the attractive potential field. The total potential field can be expressed as

$$U_{total} = \begin{cases} U_{obs} + U_{road}, |X - X_{obs}| \leq d_{ac} \\ U_{att}, else \end{cases} \tag{10}$$

where $X$ is the position of the vehicle ($x,y$). $X_{obs} = (x_{obs}, y_{obs})$, and $U_{obs}$ and $U_{road}$ represent the potential field of the obstacle and the road, respectively. The attractive potential field is

$$U_{att} = \frac{|x - x_r|}{100} + \frac{|y - 0.5D_s|}{50} \tag{11}$$

$x_r$ is the current lane centerline, and $D_s$ is the search distance.

The left and right edges of the road are defined as the road boundary potential field, which can be expressed as a function of the longitudinal length and lateral width of the road.

$$U_{road} = A_x + A_y \times A \tag{12}$$

where $A_x$ is the variation function of potential energy in the width direction, $A_y$ is the potential energy variation function in the length direction of the road, and $A$ represents the amplitude of $A_y$.

$$A_x = e^{-(|x-x_r|-x_l)^2}$$
$$A = 0.5e^{-(x-x_r)^2}$$
$$A_y = \begin{cases} 0, |y-y_{obs}| \le d_{ac} \\ \frac{|y-y_{obs}|-d_{ac}}{d_t-d_{ac}}, d_{ac} < |y-y_{obs}| \le d_t \\ 1, else \end{cases} \tag{13}$$

where $x_l$ is the centerline of the left lane, and $d_{ac}$ is the best distance that the vehicle should keep from the obstacle vehicle. When the distance between the vehicle and the obstacle is less than $d_{ac}$, the influence of the road length on the potential field function is 0. Then, the driver's reaction time $t_r$, the maximum braking force of the wheel $F_b$, the body length $L_f$, and the speed of the preceding vehicle should be considered, and the distance formula from the vehicle to the obstacle can be expressed as:

$$d_{ac} = \frac{0.5m(v_1{}^2 - v_f{}^2)}{4F_b} + t_r(v_1 - v_f) \tag{14}$$

According to Formulas (12)–(14), the 3D risk assessment map of the road potential can be obtained, as shown in Figure 5.

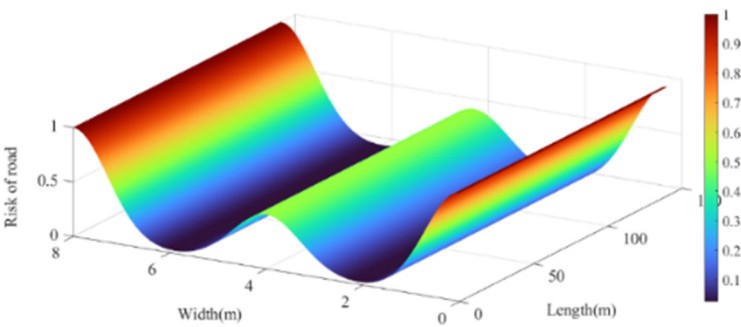

**Figure 5.** Risk of road assessment.

*3.2. Obstacle Risk Assessment Model*

The repulsive potential field of the obstacle is the key to the risk assessment of the potential field; it mainly keeps the vehicle from the obstacle at a suitable safe distance. The closer to the obstacle, the more repulsive force will be generated. On the basis of relative kinematic information, a modified two-dimensional Gaussian function is introduced to describe the collision risk, as shown in [24]

$$U_{obs} = \frac{\left|e^{-q_1(x-x_{obs})^2 - q_2(y-y_{obs})^2} - U_t\right|}{1 - U_t} \tag{15}$$

where $q_1$ and $q_2$ are the variable coefficients of the potential field in the lateral and longitudinal directions of the road. $U_t$ is the threshold of the obstacle repulsion potential field. The longitudinal coefficient $q_2$ is

$$q2 = -\frac{64F_b{}^2 \ln(U_t)}{[m(v_1{}^2 - v_f{}^2) + 4F_b L_f + 8F_b t_r(v_1 - v_f)]^2} \tag{16}$$

where $L_f$ is the length of obstacle vehicle. The longitudinal and variable coefficient prevents the vehicle from colliding with an obstacle; the horizontal and variable coefficient $q_1$ is related to the width of the road, and mainly it prevents collision with the road boundary.

That is to say, when the vehicle and the obstacle are not in the same lane, or the distance is more than the $d_{ac}$, the $q_1$ is 0.

$$q_1 = \begin{cases} \dfrac{-4[\ln(U_t)+q_2(y-y_{obs})^2]}{L_w{}^2[\sin(\frac{y-y_{obs}+d_{ac}}{d_{ac}}\times\pi-\frac{\pi}{2})+2]^2}, & |y-y_{obs}| \le d_{ac} \\ 0, else \end{cases} \tag{17}$$

According to Formulas (10)–(17), the 3D risk assessment map of the total potential can be obtained, as shown in Figure 6. The potential field at the lane centerline is the lowest, and the potential field of the obstacle is highest, which can be clearly seen. The risk value of each path is

$$path(i) = \sum_{i}^{n} U_{total}(r_i) \tag{18}$$

where $path(i)$ is the risk value of the i-th path, and $r_i$ is the i-th path point. The candidate paths are determined by the lowest risk assessment.

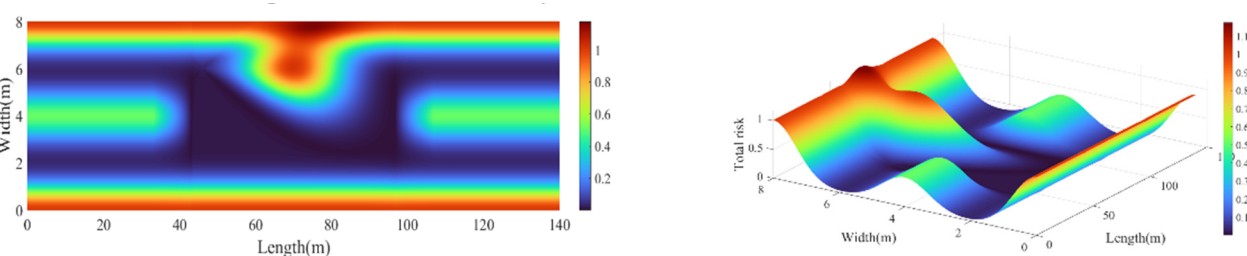

**Figure 6.** Total potential field risk assessment.

## 4. Optimal Path Selection

Safety is the most important factor in driving, so the candidate paths must be the safest paths. At the same time, the comfort of the lane change process will affect the driver's actual feeling, and the efficiency of the lane change will have an impact on the surrounding vehicles and traffic environment. Once the candidate paths have been determined, the optimal path is determined by comprehensively considering the comfort and efficiency of the lane change. The optimal path is determined by a linear weighting method, which is commonly used to transform a multi-objective optimization problem into a single objective. Thus, the total cost function is

$$\min J_{total}(i) = w_c f_c(i) + w_e f_e(i) \\ \begin{cases} w_c + w_e = 1 \\ 0 < w_c < 1 \\ 0 < w_e < 1 \end{cases} \tag{19}$$

where $w_c$ and $w_e$ are the weight value for comfort and efficiency of the lane change, $f_c(i)$ is the comfort evaluation function for the i-th path, and $f_e(i)$ is the efficiency of lane change evaluation function for the i-th path.

The efficiency of the lane change is an important indicator for the path planning; if the efficiency is too low, the time will become long, thus affecting traffic and the safety of the vehicle. Therefore, the total length of the path is selected as the efficiency cost function. In addition, it is defined as the length between the starting point $P_0$ to the end point $P_5$. $S(i)$ is the length of the path i.

$$f_e(i) = S(i) = \sqrt{(x_{P_5}-x_{P_0})^2 + (y_{P_5}-y_{P_0})^2} \tag{20}$$

Under the premise of ensuring safety, comfort is also one of the important evaluation indicators for path planning. If the curvature of the path is too high, it will not only reduce

comfort, but also increase the difficulty of path tracking control, which leads to a decrease in driving stability. The smaller the curvature change in the candidate paths, the smoother the path and the better the comfort. Therefore, the average curvature along this path as a cost function $f_c(i) = k(i)$, in [29] $k(i)$ is

$$k(i) = \frac{x'(i)y''(i) - x''(i)y'(i)}{\left[x'^2(i) + y'^2(i)\right]^{3/2}} \tag{21}$$

To reduce the difference in order of magnitude between the cost functions, the compensation function $G_a$ is introduced separately for the cost functions.

$$Ga = \begin{cases} \frac{G_a}{\max f_e(i)} \\ \frac{G_a}{\max f_c(i)} \end{cases}, \{a = f_e(i), f_c(i)\} \tag{22}$$

Comfort and efficiency of the lane change are two indicators in conflict. Aggressive-style drivers usually have higher efficiency, while not considering the comfort of the lane change, so the weight of efficiency is the highest. The path with too high risk will not be considered. For the normal-style driver, a balance should keep between lane-changing efficiency and comfort. Based on a large number of simulation experiments, the weights of the driver's lane-changing efficiency and comfort in the aggressive style are 0.7 and 0.3, respectively. The weights for the normal style are 0.5 and 0.5, respectively.

## 5. Results and Analysis

The simulation software used in this paper was MATLAB and Carsim. The simulation environment built in Carsim software includes two-lane roads, stationary vehicles, and moving vehicles. The road width is 4 m, the road boundary of the right lane is marked as 0, and the road boundary of the left lane is marked as 8. The starting point of the vehicle is (0,0). The track control of the simulation process uses the sliding mode control based on the driver preview. The preview time and other simulation parameters are shown in Table 1.

**Table 1.** Vehicle simulation parameters table.

| Parameters | Value |
|:---:|:---:|
| Sprung mass | 1370 kg |
| Speed | 60 km/h |
| Wheelbase | 2866 mm |
| Single lane width | 4000 mm |
| Obstacle vehicle length | 4600 mm |
| Potential energy threshold | 0.01 |
| Driver reaction time | 0.35 s |
| Preview time | 0.6 s |

### 5.1. The Scene of Static Obstacles

In the static scene, the position of the obstacle vehicle is (2,50) and its speed is 0 km/h. The speed of the vehicle is 60 km/h. According to Formula (18), the risk assessment value of all paths was determined, as shown in Figure 7. The first path with the lowest security is 0.48 and the 16th path with the highest risk is 0.75. Therefore, after many simulation experiments, the paths with risk assessment values less than 0.6 were selected as the candidate paths. The comfort cost value and efficiency cost value for the different paths, as shown in Figure 8a. The comfort index of the ninth path was the best, with a value of 0.717. The lane change efficiency index of the fourth path was the highest, with a value of 0.8261. The optimal lane-changing paths under different driving styles are shown in Figure 8b. The optimal path is the first path under the aggressive style, and the minimum total cost values are for the fifth path under the normal style. The comparison of the two paths during lane changing is shown in Figure 8c. The total length of the aggressive-style

path is 39.55 m, and the total length of the normal-style path is 43.52 m. The aggressive style will have shorter lane change distance and time than the normal style. The starting point and the ending point of the path are, respectively, the control points $P_0$ and $P_5$ of the fifth order Bézier curve. The comparison of the yaw angles between aggressive-style and normal-style vehicles is shown in Figure 8d. The aggressive-style angle is turned earlier, which means the steering is earlier than the normal style, and the maximum yaw angle is 6.696, which is larger than the 6.01 of the normal style.

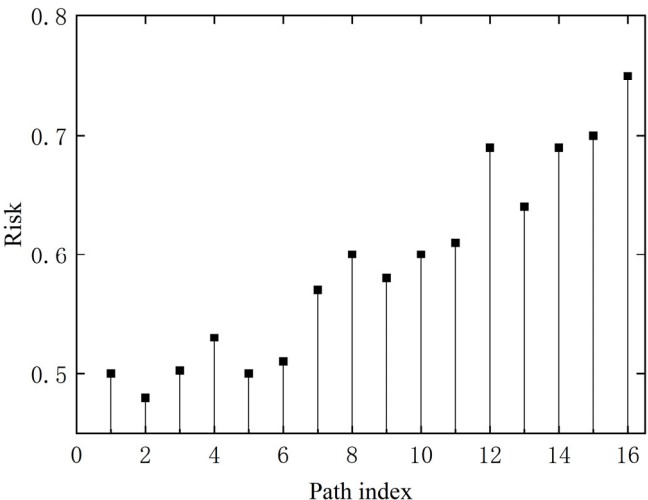

**Figure 7.** Static risk assessment values of paths.

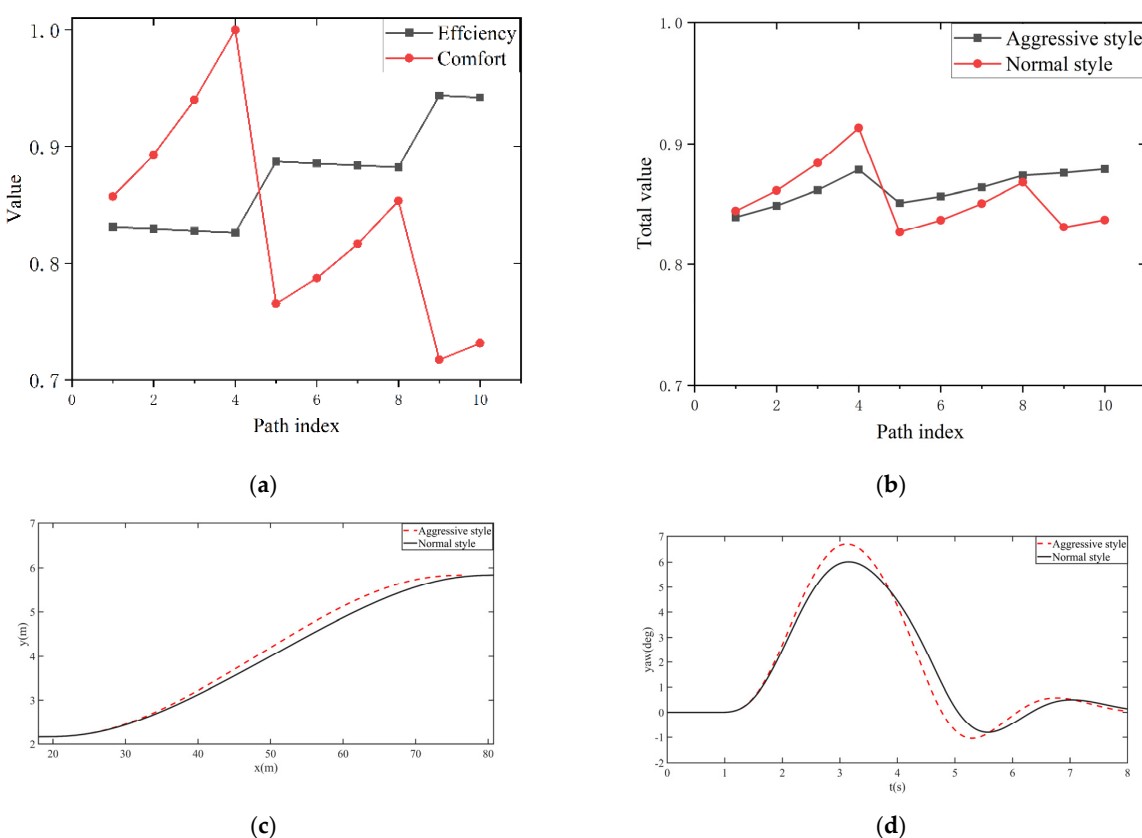

**Figure 8.** (**a**) Static cost function values of candidate paths, (**b**) the total cost values of different styles in static scene, (**c**) comparison of different style paths, and (**d**) comparison of different styles' yaw angles.

The lane change process in normal style is shown in Figure 9. It basically ends at 3.288 s and the vehicle can avoid obstacles well based on the planned optimal path.

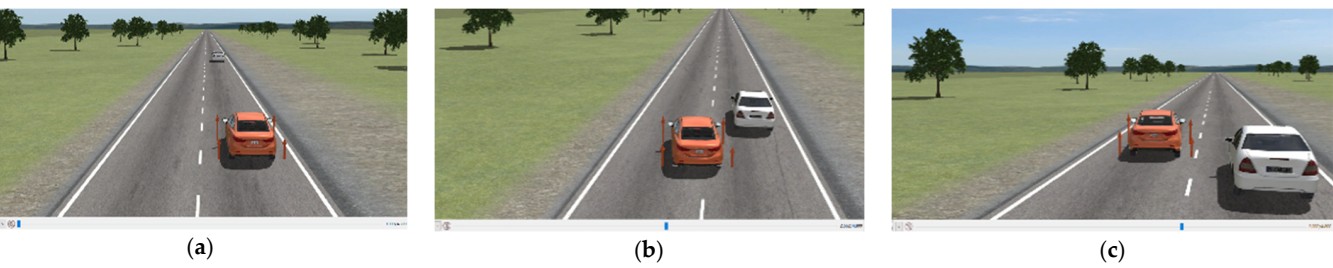

(**a**)    (**b**)    (**c**)

**Figure 9.** (**a**) Scene before lane change, (**b**) Scene during lane change, (**c**) Scene after lane change.

*5.2. The Scene of Dynamic Obstacles*

In the dynamic scene, the speed of the vehicle and the obstacle vehicle are 60 km/h and 40 km/h, respectively. The initial distance between the two vehicles is 50 m. The constraints of L1 and L2 are obtained according to Formulas (7) and (9). The dynamic risk assessment values for the paths generated under the constraints are shown in Figure 10. After many simulation experiments, the risk assessment values less than 0.65 are selected as the candidate paths. According to Formulas (19)−(22), the comfort index of the 13th path is the best, with a value of 0.7886. The lane change efficiency index of the 12th path is the highest, with a value of 0.9430. The cost values of efficiency and comfort for path candidates are shown in Figure 11a. The total cost values of the two different styles are shown in Figure 11b. The optimal path for the aggressive style was path 9, and the total length was 72.38 m. The optimal path for the normal style was path 13 and the total length was 76.36 m. The comparison of paths and yaw angles are shown in Figure 11c,d. The maximum yaw angle of the aggressive style was 5.7596, which is larger than the 5.2068 of the normal style. Compared with the scene of static obstacles, the scene of dynamic obstacles needs a longer distance to plan a safe and smooth path and a lower angle to ensure driving stability. The lateral velocity vy during the lane change is shown in Figure 11e, and the aggressive style has faster lateral speed during lane change due to the shorter path and larger yaw angle.

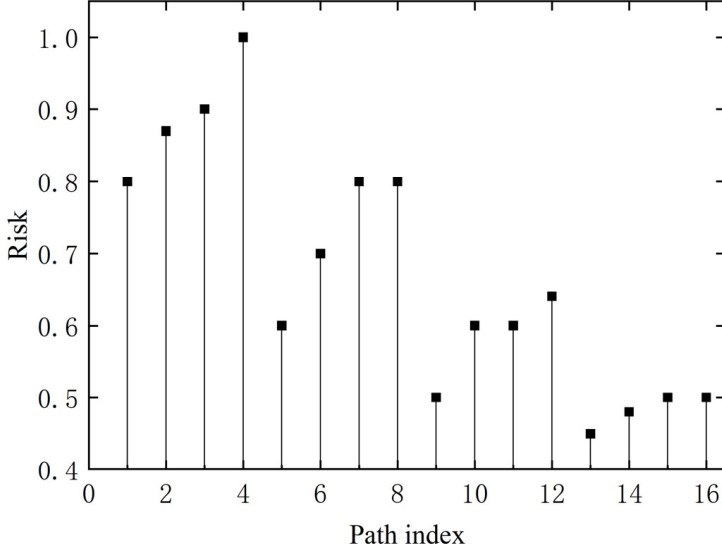

**Figure 10.** Dynamic risk assessment values of paths.

As expected, the aggressive style had greater yaw angle during lane change than the normal style. The lane change process of the aggressive style is shown in Figure 12, and Figure 12a shows the following phase before the lane change, Figure 12b shows that the

vehicle has started to change lanes, and Figure 12c clearly shows the great effect of the planned path without collision between the two vehicles during the lane change. Figure 12d shows the lane change and obstacle avoidance when they are basically over.

**Figure 11.** (**a**) Dynamic cost function values of candidate paths, (**b**) the total cost values of different styles in the dynamic scene, (**c**) comparison of different style paths, (**d**) comparison of different styles' yaw angles, and (**e**) comparison of the lateral velocity.

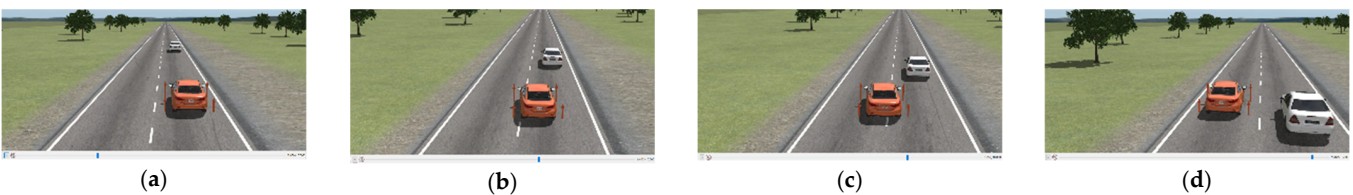

(**a**)      (**b**)      (**c**)      (**d**)

**Figure 12.** (**a**) Scene before lane change in aggressive style, (**b**) Begin lane change, (**c**) End of lane change, (**d**) Scene after lane change in aggressive style.

### 5.3. Experimental Results

Real−vehicle experiments with static scenarios were conducted on the autonomous driving vehicle platform. The platform is shown in Figure 13. The platform collects the position and motion parameters of the vehicle by combining inertial navigation CGI-410 and two directional positioning antennas. The surrounding environment can be obtained through LiDAR and camera, as shown in Figure 14. The obstacle is a stationary vehicle in the right lane. At the same time, the MPC algorithm in the Autoware open-source software was used to track and control the planned path. The speed of the vehicle was set to 5 m/s. Figure 15 shows the comparison between the real-vehicle tracking path and the desired path under the aggressive style and the normal style. The real vehicle can follow the planned path well in different styles and avoid obstacles perfectly. The maximum tracking error in the longitudinal direction was 0.1 m in normal style.

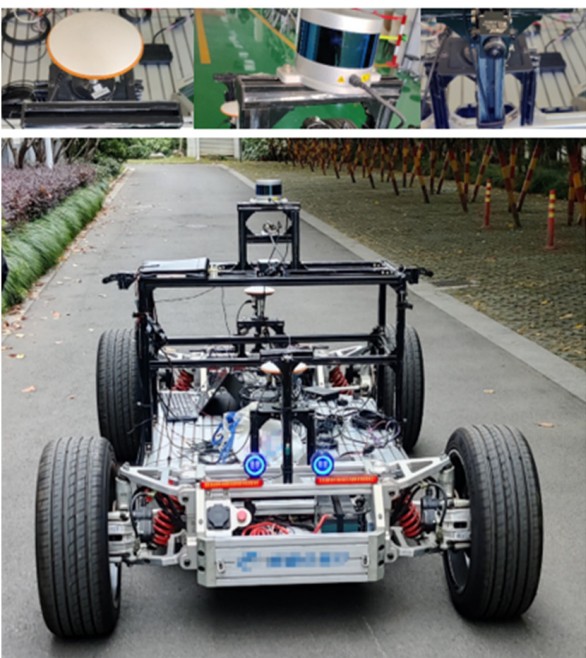

**Figure 13.** Experimental platform.

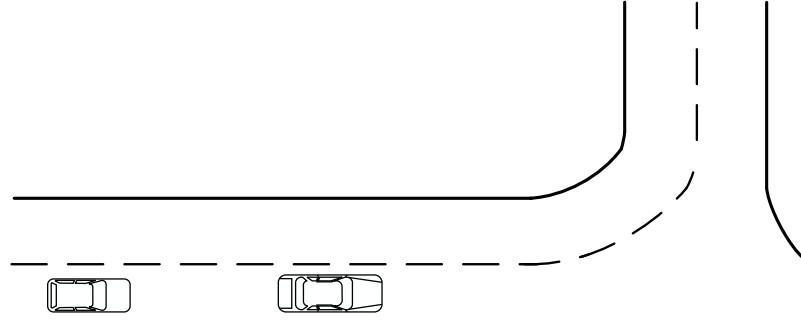

**Figure 14.** Experiment scene map.

The simulation results show that the proposed algorithm can provide different styles of optimal lane-changing paths for vehicles, and the vehicle lane-changing performance is excellent in the presence of both static and dynamic obstacles. At the same time, we verified the algorithm on the real-vehicle platform, and those results also met the requirements of lane changing and obstacle avoidance.

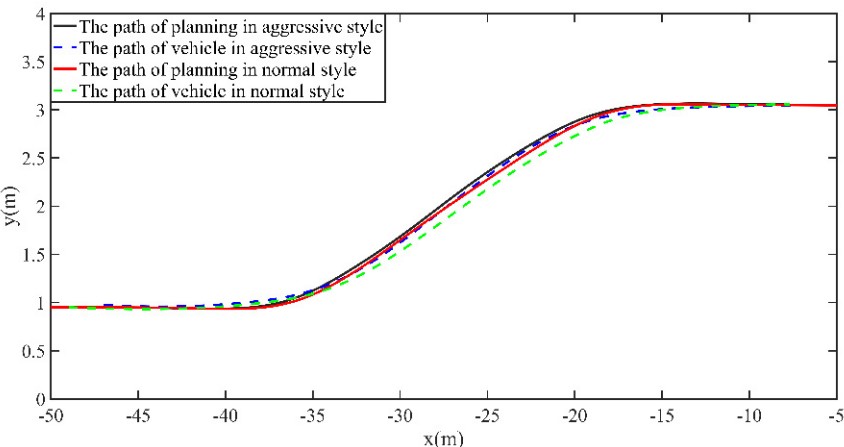

**Figure 15.** Comparison of real−vehicle experiments.

## 6. Conclusions

(1)  In this paper, the vehicle is simplified as an ellipse considering the length, width, and speed information, which makes the model more accurate in collision solution. Then, the control points of the fifth order Bézier curve are constrained to generate a series of trajectories in a safe range.

(2)  The APF model, which takes the driver's reaction time into account, conducts risk assessment on each path and selects the path most suitable for the driver's habits under the aggressive or the normal style. The results of both simulation and experiment show that the algorithm proposed in this paper has a good effect on driverless vehicles' lane changing and obstacle avoidance. In the future, continuous lane changing and obstacle avoidance under the condition of multiple obstacles will be considered, and the trajectory prediction of lane changing will be introduced.

**Author Contributions:** Review and editing, C.L.; software and validation, Y.Z.; Writing—original draft W.Y. All authors have read and agreed to the published version of the manuscript.

**Funding:** This research received no external funding.

**Data Availability Statement:** Not applicable.

**Acknowledgments:** The authors would like to thank Cong Li for critically reviewing the manuscript.

**Conflicts of Interest:** The authors declared no potential conflict of interest with respect to the research, authorship, and publication of this article.

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
