# Peer review of "A Path Planning Method for Autonomous Vehicles Based on Risk Assessment"

_wevj, doi:10.3390/wevj13120234_

Round 1

Reviewer 1 Report

The authors propose an automatic obstacle avoidance (lane change model) approach for autonomous vehicles context. The idea sounds interesting and scientific, and the main highlighted points follow such as:

1) The Abstract section could be rewritten to enhance the motivation, literature gaps, and the proposal's innovation. The text is not clear enough, so the reader has difficulty understanding the big-picture of the core idea of the paper. A brief summarization of the quantitative results is also necessary to be described. 

2) Break line 37 into a new paragraph. 

3) Section 1 appears to be written using different fonts and formats. Please check again the template used to write the article. 

4) Section 1 is a mix of an introduction to the problem under study and a literature review. The idea was good, but the discussion is centralized in papers/contributions a little outdated. ADAS system is a very traditional topic, and there are a lot of brand-new publications about this topic (2021 and 2022, mainly). The authors are invited to proceed with a significant revision in this section and clarity the innovations of the proposal when compared with very recent literature. It would be attractive to clarify different approaches under the same perspective of the target problem and place the paper's contribution in a branch of solutions found in the literature.

5) Section "1. Lane Change Curve Models" should be Section 2. The authors invested a significant part of the text in explaining equations and theories already known in the literature. This section has the potential to be drastically shortened. 

6) Same comments are valid to section "2. Risk assessment process". Indeed, that section should be the third ones. Here the paper presents several limitations when the presentation of the contributions because it is not clear if section "2. Risk assessment process" is a fundamental theory about the problem or contributions. From the reviewer's perspective, this section is only the reproduction of a view already validated in the literature. 

7) Section "2. Optimal path selection" should be the fourth section of the paper. From what is understanding, this section is considered the core and essential part of the paper. Unfortunately, it is not possible to detect any methodology or other kind of formal presentation of the main idea. Rescuing the early part of the central core idea, yet in the abstract, it would be possible to note horizon is promising. However, the authors do not present the contribution and proposal adequately. 

8) Section "2. Results and analysis" should be the fifth section of the paper. Again, a proper presentation is missing. Results are harsh to reproducible, and evaluation scenarios are mixed where it is impossible to detect where to start and finish the experimentation. Despite the figures and results being promising, it is tough to understand how they were generated. Reproducibility is a real problem in the presentation because there is no methodology that the reader can follow. 

9) Conclusions are not supported by data/results according to difficulties presented in the previous sections. Future studies are missing too. 

10) Overall, the envisaged problem under study is very interesting and opens opportunities for scientific contributions. However, the presentation of the proposal is chaotic, and there is no methodology or flow where it is possible to reproduce ideas and results. 

Reviewer 2 Report

11. The formatting of the text is not clear, so it is not clear whether the article is "glued together" from separate pieces or it is one piece

  2. The given formulas are not linked, so it is not clear whether they are created by the author or taken from sources, although most of them are known

33. The article has three second parts, so the structure of the article is not clear

44. The article uses the probabilistic method of ellipses, but it does not reflect what the probabilities of collision or non-collision of vehicles are

5 5. Formulas (15) and (16) are not clear, since the values of different dimensions are added

6 6.  In the theoretical part (second chapters), graphic processes are not displayed, so the presented formulas will be difficult for readers to understand

7 7. The article does not mention the relationship "machine - man", which is the biggest problem, because the relationship "machine - machine" is, for example, successfully solved in autonomous container terminals

8 8. It is recommended that the article be substantially restructured to make it clear, as the article makes its goals very clear.

1

Reviewer 3 Report

Authors present a paper about a path planning method for autonomous vehicles based on risk assessment. 

The area of autonomous cars is an increasingly interesting and pressing area as electric cars evolve (mainly) and the intention to evolve towards autonomous cars to reduce accidents, for example. And so this paper is very interesting.

However, I don't understand why the lack of care in formatting the paper.

For example, we found the text with different font sizes several times, we found the equations and figures out of place, we found the wrong referencing standard, etc...

It is advisable to reformat and correct all the mistakes that the paper has and that in some cases even make it difficult to understand the work done.

Round 2

Reviewer 1 Report

The previous mentioned highlights were improved. Thus, the reviewer recommendations it to approve the paper in the present forms. 

Author Response

Thank you very much for your comments,Please see the attachment.

Reviewer 2 Report

1. Image 14 is difficult to read, it is recommended to replace it with a better quality one.

2. It is recommended to provide at least a short summary at the end of Chapter 5 (after Figure 15).

Author Response

Thank you for your comments. We have made changes according to your comments. Please see the attachment
